# *In vivo* nanoscale analysis of the dynamic synergistic interaction of *Bacillus thuringiensis* Cry11Aa and Cyt1Aa toxins in *Aedes aegypti*

Samira López-Molina[1], Nathaly Alexandre do Nascimento[2], Maria Helena Neves Lobo Silva-Filha[2], Adán Guerrero[3], Jorge Sánchez[1], Sabino Pacheco[1], Sarjeet S. Gill[4], Mario Soberón[1], Alejandra Bravo[1] *

1 Departamento de Microbiología Molecular, Instituto de Biotecnología, Universidad Nacional Autónoma de México (UNAM), Cuernavaca, Morelos, Mexico, 2 Department of Entomology, Instituto Aggeu Magalhães-FIOCRUZ, Recife-PE, Brazil, 3 Laboratorio Nacional de Microscopía Avanzada, Instituto de Biotecnología, UNAM, Cuernavaca, Morelos, Mexico, 4 Department of Molecular, Cell and Systems Biology, University of California, Riverside, Riverside, California, United States of America

* bravo@ibt.unam.mx

**Data Availability Statement:** All relevant data are within the manuscript and its Supporting Information files.

## Abstract

The insecticidal Cry11Aa and Cyt1Aa proteins are produced by *Bacillus thuringiensis* as crystal inclusions. They work synergistically inducing high toxicity against mosquito larvae. It was proposed that these crystal inclusions are rapidly solubilized and activated in the gut lumen, followed by pore formation in midgut cells killing the larvae. In addition, Cyt1Aa functions as a Cry11Aa binding receptor, inducing Cry11Aa oligomerization and membrane insertion. Here, we used fluorescent labeled crystals, protoxins or activated toxins for *in vivo* localization at nano-scale resolution. We show that after larvae were fed solubilized proteins, these proteins were not accumulated inside the gut and larvae were not killed. In contrast, if larvae were fed soluble non-toxic mutant proteins, these proteins were found inside the gut bound to gut-microvilli. Only feeding with crystal inclusions resulted in high larval mortality, suggesting that they have a role for an optimal intoxication process. At the macroscopic level, Cry11Aa completely degraded the gastric *caeca* structure and, in the presence of Cyt1Aa, this effect was observed at lower toxin-concentrations and at shorter periods. The labeled Cry11Aa crystal protein, after midgut processing, binds to the gastric *caeca* and posterior midgut regions, and also to anterior and medium regions where it is internalized in ordered "net like" structures, leading finally to cell break down. During synergism both Cry11Aa and Cyt1Aa toxins showed a dynamic layered array at the surface of apical microvilli, where Cry11Aa is localized in the lower layer closer to the cell cytoplasm, and Cyt1Aa is layered over Cry11Aa. This array depends on the pore formation activity of Cry11Aa, since the non-toxic mutant Cry11Aa-E97A, which is unable to oligomerize, inverted this array. Internalization of Cry11Aa was also observed during synergism. These data indicate that the mechanism of action of Cry11Aa is more complex than previously anticipated, and may involve additional steps besides pore-formation activity.

**Funding:** This work was supported by Dirección General de Asuntos del Personal Académico de la Universidad Nacional Autónoma de México-DGAPA/UNAM (https://dgapa.unam.mx) grant number: IN202718 (to SP) and DGAPA/UNAM grant number: IN203619 (to AB) and National Institutes of Health-NIH 2R01 AI066014 (to SSG) https://www.nih.gov. The funders had no role in study design, data collection and analysis, decision to publish, or preparation of the manuscript.

**Competing interests:** The authors have declared that no competing interest exist in this work.

## Author summary

It was proposed that Cyt1Aa functions as a receptor of Cry11Aa, explaining their synergism that increases toxicity against mosquitos. *In vivo* interaction of Cry11Aa and Cyt1Aa proteins has never been studied. Here, we describe the effects of both toxins on gut structures, their localization and action after ingestion. We use quantitative super resolution imaging to analyze their interaction, showing a dynamic and highly ordered array of these proteins in the apical microvilli of midgut cells and internalization of Cry11Aa upon pore formation activity. Our data indicate that the mechanism of action of these pore-forming toxins differs, since Cyt1Aa exerts its activity on the plasma membrane, while additional intracellular effectors are likely to be involved in toxicity of Cry11Aa toxin.

## Introduction

Bacterial pathogens make use of different and diverse virulent factors to infect their hosts and disrupt target cells. Among these factors are pore forming toxins that are highly efficient. *Bacillus thuringiensis* (Bt) bacteria are insect pathogens that produce, among other virulence factors, different pore forming toxins to break down the midgut epithelial cells of their target insects [1]. During the sporulation phase of growth, different Bt strains produce diverse δ-endotoxins that are concentrated in the mother cell as crystal inclusions [1]. The insecticidal δ-endotoxin proteins characterized so far, are pore forming toxins named Cry or Cyt proteins. The insecticidal Cry proteins consist of at least three different non-related phylogenetically families showing different primary sequences and three-dimensional structures [2].

Most Bt strains that kill mosquitoes produce more than one Cry and Cyt toxins that contribute to the toxicity against the target insect [1,2]. One case is Bt var *israelensis* (Bti) that produce at least three Cry toxins, such as Cry4Aa, Cry4Ba, and Cry11Aa and one major Cyt toxin, the Cyt1Aa [3,4]. Bti has a spectrum of action against dipteran insects including *Aedes aegypti*, *Anopheles gambiae* and *Culex pipiens* mosquitoes, which are vectors of several important human diseases such as dengue, yellow fever, Zika, chikungunya, malaria, encephalitis and filariasis [3,4]. Both, Cry and Cyt proteins, are produced as protoxins that accumulate as crystal inclusions during Bt sporulation. Mosquito larvae live in an aquatic environment, where they feed by filtration of particles including bacteria, zooplankton and algae. It was estimated that the particle size filtered by mosquitoes ranged from 0.7 to 26 μm [5], and that the crystal inclusions of Bti have an average size of 1 μm [6,7]. When larvae ingest the crystal inclusions, they are solubilized in the alkaline and reducing conditions of the gut and these soluble protoxins are then activated by midgut proteases [1,4]. Specifically, the Cry11Aa protoxin of 70 kDa is activated into two fragments of 32 and 36 kDa that remain associated as a single ≈65 kDa activated toxin. The Cry activated toxins are composed of three domains, where domain I is a seven α-helix bundle involved in toxin oligomerization, membrane insertion and pore-formation. While domains II and III, mainly composed of β-strands, which are involved in specific interactions with protein receptors, and thus directly participate in toxin specificity [1,4]. In contrast, the activated Cyt toxin has a size of 23 kDa, and it is composed of a single α-β domain, where the two outer layers of α-helix hairpins wrap around a long β-sheet [4,8–10].

It was previously reported that solubilized Bti crystal proteins were 7000-fold less efficient in killing mosquito larvae than their crystal inclusions [11]. Also, soluble toxin proteins adsorbed into 0.8 μm latex beads recovered high toxicity [11]. Moreover, larvae survive when fed with soluble Cry11Aa or Cyt1Aa proteins, in contrast to their crystal inclusions that showed high toxicity [12].

The mosquitocidal Cry toxins rely on binding to different larval midgut proteins such as alkaline phosphatase (ALP), aminopeptidase-N (APN) or cadherin (CAD) for their membrane insertion and pore formation. These host proteins function as toxin receptors triggering oligomerization and membrane insertion of the toxin, leading to pore formation and finally to the cell lysis [1]. In contrast, the Cyt toxin does not rely on binding to gut protein receptors, but it specifically binds to lipid membranes suggesting a detergent-like mode of action, although it was also shown that binding to lipids triggers toxin oligomerization, insertion into the membrane and pore formation [1,13].

Bti has been widely used for the control of different dipteran pests in the field for more than forty years, without evidence of resistance evolution by the target insects; neither has it been possible to artificially select resistant mosquitoes to Bti crystal inclusions under laboratory conditions [14]. Interestingly, the lack of resistance toward Bti inclusions was shown to be mainly due to the presence of the Cyt1Aa protein. Resistant insects towards each of the three individual Cry4Aa, Cry4Ba and Cry11Aa toxins have been selected in laboratory conditions, but no resistant population could be selected against the Cyt1Aa protein [14]. In addition, it was shown that in the presence of Cyt1Aa, the Cry-resistant larvae recovered their susceptibility to these Cry4Aa, Cry4Ba and Cry11Aa proteins [15]. Moreover, Cyt1Aa synergizes the toxicity of Cry11Aa and Cry4Ba to different mosquito larvae, acting as their receptor, which greatly reduces the selection of resistance [16–20]. It was shown that Cyt1Aa and Cry11Aa proteins bind each other in a high affinity interaction, that directly correlates with their synergism, as demonstrated by point mutations in the specific regions of Cry11Aa and Cyt1Aa that participate in their binding interaction [17]. Specifically, the exposed domain II loop regions of both Cry11Aa and Cry4Ba toxins, which are involved in the interaction with larval gut ALP receptor, also contribute to their binding to Cyt1Aa [17,19]. Furthermore, it was also shown that Cyt1Aa induce oligomerization of Cry11Aa resulting in pore formation in membrane vesicles [18], showing that Cyt1Aa functions as a receptor of Cry11Aa and Cry4Ba proteins. These data explain its capacity to counter resistance to Cry toxins and to synergize their toxicity against mosquitoes [17–19].

Despite the fact that the mode of action and the synergistic mechanism between Cyt1Aa and Cry11Aa has been described in some detail, their localization inside the gut after ingestion and the effects of both toxins on the midgut cells of the gut structures have not been studied. Immunocytochemical localization of the individual toxins of Bti, after *in vitro* binding to midgut tissue isolated from *An*. *gambiae* or *Ae*. *aegypti* larvae, showed that Cry11Aa mainly binds to brush border membrane (BBM) of the gastric *caeca* and posterior midgut regions. While Cyt1Aa binds to BBM from all gut regions including the microvilli of cells from the *caeca*, and the anterior and posterior midgut regions [21,22]. The binding of Cry11Aa to microvilli from the gastric *caeca* and posterior midgut correlates with the localization of ALP, APN and CAD receptors in these midgut regions [22–25]. Nevertheless, the interaction of Cry11Aa and Cyt1Aa proteins has never been studied *in vivo*.

## Results

### Comparative activity of Cry11Aa and Cyt1Aa proteins against *Aedes aegypti* larvae

We compared the toxicity of crystal inclusions and soluble proteins of Cry11Aa and Cyt1Aa. As controls we used two non-toxic mutants: Cry11Aa-E97A and Cyt1Aa-V122E. The former protein contains a point mutation at domain I of Cry11Aa, and the later contains a point mutation within helix α-3 of Cyt1Aa. Both mutants exhibit very low toxicity because oligomerization is affected [26,27]. Fig 1 shows that crystal inclusions of Cry11Aa were highly toxic to

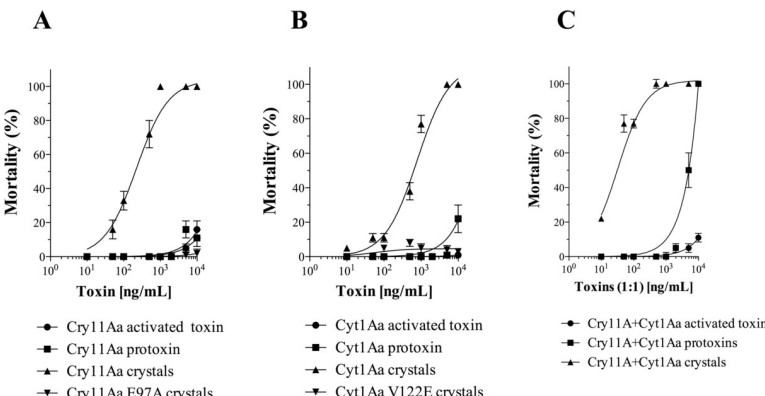

**Fig 1. Insecticidal activity of soluble activated toxin, soluble protoxin and crystal inclusions from Cry11Aa and Cyt1Aa, wild type or mutant proteins against 3rd instar *Aedes aegypti* larvae after 24 h of exposure.** Each bioassay was repeated three times with five replicates. The $LC_{50}$ values and 95% confidence intervals (CI) of the samples to larvae were calculated by using Polo Plus Probit and Logit Analysis version 1.0 LeOra software, as follows: Cry11Aa crystals = 329.3 (CI: 203.1–605.6) ng/ml; Cyt1Aa crystals = 679.23 (CI: 554.9–851.1) ng/ml; Cry11Aa- Cyt1Aa 1:1 crystals mixture = 51.53 (CI: 554.9–851) ng/ml; soluble Cry11Aa -Cyt1Aa 1:1 protoxins mixture = 5335 (CI: 4503–6339) ng/ml. $LC_{50}$ could not be calculated for those samples without larvicidal activity.

3rd instar larvae, showing a medium lethal concentration ($LC_{50}$) value of 329.3 ng/ml (95% confidence intervals (CI) = 203.1–605.6). In contrast the crystal inclusions of the negative control Cry11Aa-E97A mutant protein did not show any toxicity even at 10,000 ng/ml, as previously reported [26]. When larvae were fed with soluble Cry11Aa protoxin or activated toxin only low mortality was observed in the larvae at the highest tested concentration (10,000 ng/ml) (Fig 1A). In the case of Cyt1Aa, similar results were obtained, Cyt1Aa crystal inclusions were toxic to larvae ($LC_{50}$ = 679.23 ng/ml, 95% CI: 554.9–851.1), while crystal inclusions from the negative control Cyt1Aa-V122E mutant showed negligible toxicity, as expected (Fig 1B) [27]. When larvae were fed soluble Cyt1Aa activated toxin or protoxin, negligible mortality of the larvae was observed at 10,000 ng/ml (Fig 1B).

We also analyzed the toxicity of a 1:1 mixture of Cry11Aa with Cyt1Aa proteins. The mixture of Cyt1Aa and Cry11Aa crystal inclusions showed a synergism factor (SF) of 8.6 ($LC_{50}$ = 51.53 ng/ml, 95% CI: 554.9–851) according to Tabashnik's equation [28] (Fig 1C). In contrast, when larvae were fed with mixtures of soluble and activated Cry11Aa and Cyt1Aa toxins, most of the larvae survived after 24 h, showing marginal mortality (Fig 1C). While a mixture of soluble protoxins showed 100-fold lower toxicity ($LC_{50}$ = 5335 ng/ml, 95% CI: 4503–6339) compared to a mixture of crystal inclusions (Fig 1C).

## Gut integrity was severely affected after intoxication with Cry11Aa or with a mixture of Cry11Aa and Cyt1Aa proteins

To determine the effect of Cry11Aa and Cyt1Aa proteins on the larval gut structure, we fed 4th instar larvae with crystal inclusions of Cry11Aa, Cyt1Aa or a mixture of crystal inclusions from both proteins for different times, and their dissected guts were analyzed by optical microscopy.

Representative images of larval guts are shown in Fig 2. The non-treated control larva is shown in Fig 2A. The gastric *caeca* region is composed of eight blind ending tubes, which empty into the anterior end of the midgut [29]. S1 Fig shows a diagram of the larval gut structure. The images obtained after treatment with Cry11Aa crystal inclusions clearly showed that, after 24 h exposure to a high dose (1000 ng/ml), the gastric *caeca* region was completely

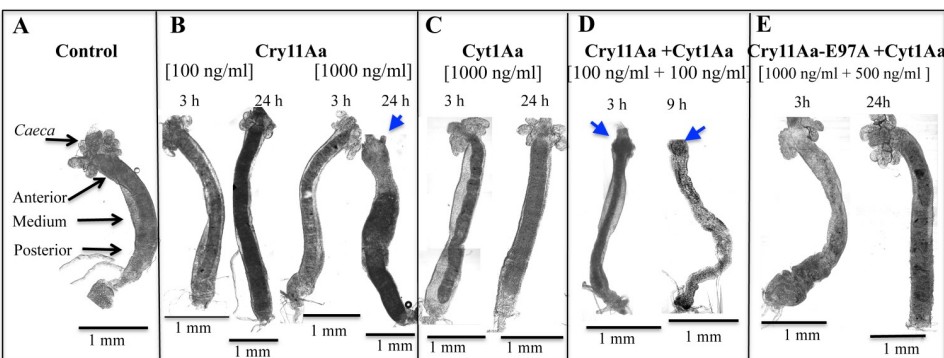

**Fig 2. Representative images of dissected midgut tissues from Cry11Aa and/or Cyt1Aa-treated** *Aedes aegypti* **4**[th] **instar larvae recorded by clear field microscopy.** Blue arrowheads point to the severely affected gastric *caeca* region of some midguts. Representative composite images were constructed with Fiji-Image J software. A total of five larvae were dissected for each condition and each assay was performed at least three times.

degraded (Fig 2B, see blue arrow). In contrast, larvae fed with Cyt1Aa crystals at that same concentration and exposure time, showed no evident changes on the *caeca*, or any other gut structure at the macroscopic level (Fig 2C), even though the tested protein concentration was over the LC$_{50}$ value and that half of those larvae died after 24 h of toxin exposure.

When larvae were fed with a mixture of Cry11Aa and Cyt1Aa crystals at ten-fold lower concentration (100 ng/ml Cry11Aa + 100 ng/ml Cyt1Aa) the *caeca* was completely degraded after only 3 h (Fig 2D, see blue arrow), probably due to the enhanced mortality observed in the bioassays performed with a mixture of both proteins (Fig 1C). After 9 h of ingestion of the mixture of crystal inclusions of Cry11Aa + Cyt1Aa, the effects in the *caeca* were more evident showing complete destruction, while the whole midgut appeared severely affected (Fig 2D). In contrast, larvae fed with a mixture of crystal inclusions from the nontoxic mutant Cry11Aa-E97A with wild type Cyt1Aa (1000 ng/ml Cry11Aa-E97A + 500 ng/ml Cyt1Aa) showed an intact *caeca* after 24 h (Fig 2E), suggesting that degradation of the *caeca* in the mixture of toxins may be due to enhanced Cry11Aa toxicity induced by Cyt1Aa.

## Soluble protoxin or activated Cry11Aa toxin were weakly observed inside the larval gut

To determine the fate of Cry11Aa or Cyt1Aa in the gut tissue of larvae fed Cry11Aa or Cry11Aa-E97A inclusions or soluble proteins were labeled with Alexa-546 (green color), while Cyt1Aa or Cyt1Aa-V122E inclusions or soluble proteins were labeled with Alexa-647 (red color) (S2 Fig). The efficiency of labeling of each protein sample was determined with respect to its molar extinction coefficient as described in Materials and methods and labeled proteins were visualized directly on the SDS-PAGE excited with an Epi-RGB trans-illuminator showing that the corresponding proteins were labeled (S2 Fig). In the case of crystal inclusions, the efficiency of labeling was significantly lower than that of soluble toxins probably due to reduced labeling of protoxins inside the crystal inclusions, although the corresponding protein bands in SDS-PAGE confirmed the labeling of these proteins (S2 Fig).

When larvae were fed activated Cry11Aa-Alexa546 soluble toxin a discrete detection of the soluble toxin was recorded associated with the microvilli membrane of the posterior midgut after 3 h of exposure (S3A Fig, see red arrowhead). However, this localization of the labeled toxin was not further observed at other intoxication times, i.e. after 6 h or longer periods of larval exposure to the toxin (Figs 3A and S3A). The concentration of the toxin inside the gut was

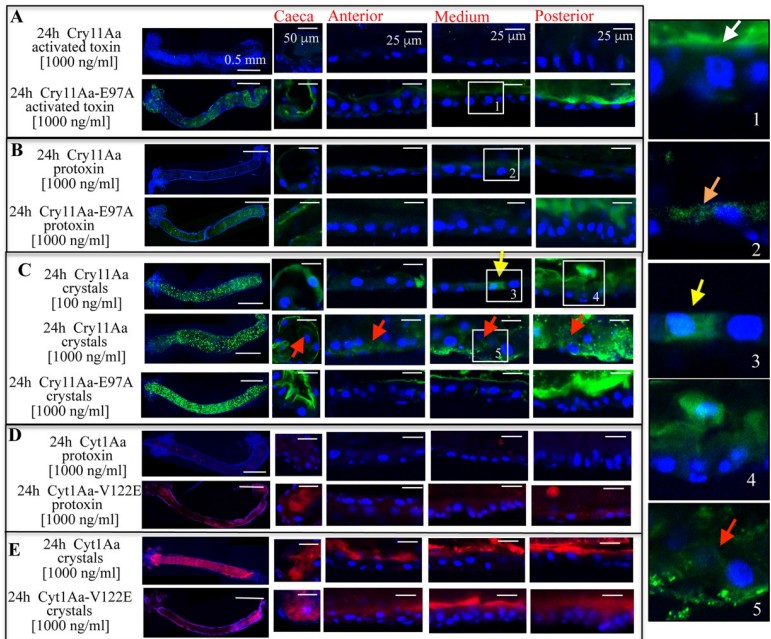

**Fig 3. *In vivo* localization of labeled Cry11Aa and Cyt1Aa, or mutant Cry11Aa-E97A and Cyt1Aa-V122E proteins, inside the midgut tissue of *Aedes aegypti* larvae.** Cry11Aa proteins were labeled with Alexa546 showing green fluorescence. Cyt1Aa proteins were labeled with Alexa647 showing red fluorescence. Individuals were fed with of each labeled protein (100 ng/ml or 1000 ng/ml) administrated as soluble activated toxin, soluble protoxin or crystal inclusions for 24 h. A total of five larvae were dissected for each condition and each assay was performed at least three times. Representative images are shown. Midgut tissue was dissected and processed as indicated in Materials and methods. Nucleus was stained with DAPI (showing blue fluorescence) and labeled proteins were observed with confocal laser Olympus FV1000 scanning microscope. Insets show selected images to improve clarity of the images. White arrowhead points to Cry11Aa-E97A activated toxin protein bound to microvilli membrane from medium tissue region. Orange arrowhead points to Cry11Aa protoxin internalized into small vesicles. Yellow arrowheads point to cells of the medium midgut highly brilliant due to Cry11Aa internalization. Red arrowheads point to cells that were already broken down.

marginal since after 24 h of exposure, this protein was barely observed inside the gut lumen, and it was not found associated to BBM from all midgut regions (Fig 3A).

When larvae were fed with Cry11Aa-Alexa546 soluble protoxins for 24 h, or shorter incubation times, this protoxin was also barely observed in the gut lumen (Figs 3B and S4A). After 9 h and 24 h of ingestion, some Cry11Aa-Alexa546 protoxin was found internalized in small vesicles in cells of the medium midgut region (Figs 3B and S4A, see the orange arrowheads, in the inset images).

## The soluble activated toxin or protoxin from the nontoxic mutant Cry11Aa-E97A were accumulated in the larval gut

In stark contrast, at all times analyzed, soluble activated toxin of the nontoxic mutant Cry11Aa-E97A-Alexa546 accumulated in the gut lumen, and it was found principally associated to the microvilli membranes in the gastric *caeca* and posterior midgut (Figs 3A and S3B). This protein was also observed slightly bound to microvilli membranes from the anterior and medium region (Figs 3A and S3B, see white arrowhead in the inset-1 of Fig 3A). The mutation of Cry11Aa-E97A is located in helix α-3 of domain I that is involved in oligomerization and pore formation activity, while the toxin-binding regions of this toxin are located in domains II and III, implying that binding interaction of this toxin with its receptors is not affected [26].

Similarly, at all times analyzed, the soluble protoxin from the non-toxic Cry11Aa-E97A-Alexa546 mutant was observed inside the gut lumen, and also associated with microvilli membranes, principally from the gastric *caeca* and posterior midgut regions (Figs 3B and S4B).

## Crystal inclusions from Cry11Aa or Cry11Aa-E97A were ingest by the larvae and accumulated inside the gut

Recently, it was shown that when *Ae. aegypti* larvae were fed with low doses of the Cry11Aa crystals, some regions of the microvilli were severely affected, deformed and detached from the cells, and this shedding of membranes was proposed to be a defense mechanism against Cry toxin action [25]. Here we observed that at low dose (100 ng/ml) Cry11Aa-Alexa546 crystal inclusions, were readily ingested and accumulated into the midgut lumen and the labeled protein was found bound to the *caeca* and the posterior part of the midgut (Fig 3C). The microvilli membrane from the posterior region was deformed and membrane shedding was observed (see inset-4 in Fig 3C), similar to that previously reported [25]. However, we also observed that Cry11Aa-Alexa546 was internalized in some cells of the medium midgut and in the *caeca* (Fig 3C, see yellow arrowhead and inset-3). Cell internalization was observed after 9 h of intoxication with a low dose (100 ng/ml) of Cry11Aa-Alexa546 crystals (Fig 4A, see yellow arrowhead). Larvae intoxicated with a high dose of Cry11Aa-Alexa546 crystals (1000 ng/ml) showed that all cells from their gut tissue were severely damaged after 24 h (Fig 3C, see red arrow heads and inset-5). At 3 and 6 h incubation with a high dose (1000 ng/ml) of Cry11Aa-Alexa546 crystals, the labeled protein was found associated with membranes from all midgut regions including anterior and medium midgut (Fig 4A). The internalization of this protein in certain cells of the gut was more evident after 3 h and 6 h (Fig 4A, see yellow arrows). After 9 h, cells from all gut regions were completely broken down (Fig 4A, see red arrows), and the macroscopic structure of the gastric *caeca* was clearly damaged (Fig 4A, see blue arrowhead). Fig 4C shows a lower magnification of the midgut tissue from larvae that were fed 3 h with 1000 ng/ml. Here some midgut cells showed toxin bound to the microvilli and some cells showed a strong fluorescence signal inside the cells (Fig 4C, yellow arrow heads pointed to the highly brilliant cells where Cry11Aa-Alexa546 was internalized). It is possible that in Cry11Aa-brilliant cells, the apical membrane may already be permeable, allowing Cry11Aa internalization.

In contrast, when larvae were fed with crystals of the mutant protein Cry11Aa-E97A-Alexa546 mutant at high dose (1000 ng/ml) for 24 h, this labeled protein was found in the midgut lumen and also associated to the microvilli membranes, principally from gastric *caeca* and posterior midgut. But no apparent cell damage or cell internalization was observed (Fig 3C). Similar images were observed at shorter incubation times (3, 6 and 9 h) (Fig 4B). Slight binding of Cry11Aa-E97A-Alexa546 with microvilli membrane of the anterior and medium midgut regions was also observed (Figs 3C and 4B).

The intensity of fluorescence of the cytoplasm region was analyzed in individual cells after 3 h of intoxication with 1000 ng/ml Cry11Aa-Alexa546 or with Cry11Aa-E97A-Alexa546 crystal inclusions (Fig 4C). Cytoplasmic fluorescence was measured in dozens of individual cells from each condition, including the highly brilliant (Fig 4C, see yellow arrows), the dark cells after intoxication with Cry11Aa, and cells from the larvae intoxicated with the mutant Cry11Aa-E97A-Alexa546 protein (Fig 4C). These fluorescence measurements showed that brighter cells after Cry11Aa intoxication were significantly different from darker cells, or from the midgut cells after intoxication with Cry11Aa-E97A-Alexa546 ($P$ value $< 0.001$, multiple comparison after Kruskal-Wallis test (Fig 4C).

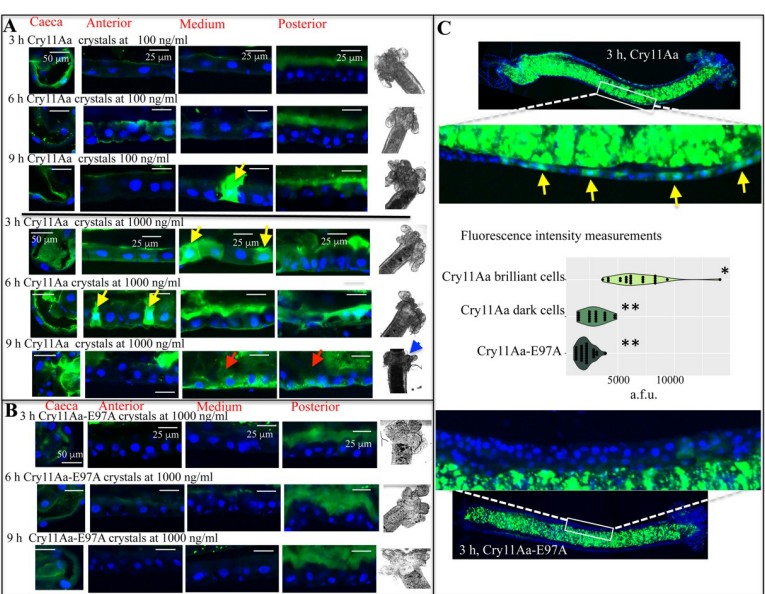

**Fig 4.** *In vivo* localization of labeled crystal inclusions from Cry11Aa wild type (Panel A) or Cry11AaE97A mutant (Panel B) in the midgut tissue of *Aedes aegypti* larvae. Individuals were fed (100 ng/ml or 1000 ng/ml) of each labeled crystal inclusions for 3, 6 and 9 h. Cry11Aa proteins were labeled with Alexa546 showing green fluorescence. A total of five larvae were dissected for each condition and each assay was performed at least three times. Representative images are shown. Midgut tissue was dissected and processed as indicated in Materials and methods. Nucleus was stained with DAPI (showing blue fluorescence) and labeled proteins were observed with confocal laser Olympus FV1000 scanning microscope. Clear field microscopy images of *caeca* region are also shown in panels A and B of this figure. Yellow arrowheads point to highly brilliant midgut cells due to Cry11Aa-Alexa546 internalization. Red arrowheads point to cells that were already broken down and the blue arrow points to the gastric *caeca* region that was severely affected. Panel C shows the complete midgut tissue and a fragment from the medium midgut region showing the highly brilliant cells and dark cells after feeding with Cry11Aa-Alexa546 crystal inclusions. Yellow arrowheads point to Cry11Aa-brilliant cells. Panel C also shows the intensity of fluorescence measurements of 20 x 20 μm squares selected from the cytoplasm of different cells from all conditions. Data are shown as arbitrary fluorescent units (a.f.u.). Different asterisks indicate statistically significant different data analyzed by using Kruskal-Wallis test ($P$ value < 0.001).

During all times of intoxication, the Cry11Aa labeled crystals were observed inside the gut lumen of the larvae (Figs 3 and 4), indicating that *in vivo* the crystal inclusions may spend a long period of time inside the larval gut.

## Soluble Cyt1Aa protoxin was not accumulated inside the larval gut and only crystal-inclusions of Cyt1Aa were found inside the larval gut

Regarding Cyt1Aa, when larvae were fed with Cyt1Aa-Alexa647 soluble protoxin, we observed marginal concentration of Cyt1Aa inside the gut lumen and the toxin was not associated with membranes from any gut regions at all times analyzed including after 24 h (Figs 3D and S5A). In contrast, the soluble protoxin from the nontoxic mutant Cyt1Aa-V122E-Alexa647 was found inside the gut lumen (Figs 3D and S5B). In the case of larvae intoxicated with Cyt1Aa-Alexa647 inclusions, confocal microscopy revealed that labeled crystals were highly accumulated in the gut lumen and Cyt1Aa-Alexa647 was associated with BBM from all gut regions, although no clear cell damage was observed (Figs 3E and S6A). Similarly, when larvae were fed with crystals of the non-toxic mutant Cyt1Aa-V122E-Alexa647 strong accumulation of this protein was observed in the gut lumen and was found associated with BBM from all gut regions (Figs 3E and S6B).

In supplementary S3–S6 Figs, bright field images of the gastric *caeca* at all periods of intoxications with the different protein samples (Cry11Aa and Cyt1Aa activated and protoxins) are shown. These images indicate that none of these conditions impaired the *caeca* structure; the only exceptions were with Cry11Aa crystals, where the damage of *caeca* structure was evident after 9 and 24 h treatment (see blue arrowheads in Figs 2B and 4A).

## Cry11Aa and Cyt1Aa colocalized during their synergistic interaction in *Aedes aegypti* larval guts

To determine the fate of Cry11Aa and Cyt1Aa proteins during their synergistic interaction in the gut, 4th instar larvae were fed with a mixture of crystal inclusions of Cry11Aa-Alexa546 and Cyt1Aa-Alexa647 (1:1 ratio, at 100 ng/ml concentration of each toxin). After 3 h of ingestion the midgut tissue was dissected and observed under the confocal microscope. Fig 5A shows that, both Cry11Aa-Alexa546 and Cyt1Aa-Alexa647 inclusions accumulated in gut lumen. Both toxins were associated and colocalized at microvilli membranes from all gut regions, including the anterior, medium and posterior midgut regions (Fig 5A). The *caeca* was already disrupted under these conditions, thus we observed no colocalization in this region. Colocalization of both toxins in the microvilli of gastric *caeca* was clearly recorded when larvae were fed with a mixture of mutant Cry11AaE97A-Alexa546 and Cyt1Aa-Alexa647 (Fig 5B).

During synergism of Cry11Aa and Cyt1Aa proteins, cell internalization of Cry11Aa-Alexa546 toxin was observed in the cells from the medium and posterior parts of the gut (Fig 5A see yellow arrows in inset-1 and inset-2). The cells from the posterior region were highly affected after 3 h of intoxication, showing severe effects in their morphology (Fig 5A, see inset-1).

When a mixture of crystal inclusions of the mutant Cry11Aa-E97A-Alexa546 and Cyt1Aa-Alexa647 (1000 ng/ml + 500 ng/ml, respectively) was used to feed larvae, both toxins were found to be associated and colocalized, and bound to microvilli membranes from all gut regions. No internalization of these toxins was evident, neither apparent cell damage was observed at 3 h (Fig 5B and inset-3) nor at 9 h (S7 Fig) of ingestion. At such times, microvilli structures are intact and the size of the cells did not change even though the concentration of labeled proteins used was much higher than that at which the wild type toxins were analyzed, as noted above (Figs 5B and S7).

Fluorescence analysis of the cytoplasm region from dozens of cells in each condition confirmed these data, since cells intoxicated simultaneously with Cry11Aa-Alexa546 + Cyt1Aa-Alexa647 have statistically significant higher fluorescence values, compared to cells intoxicated with a mixture of Cry11Aa-E97A-Alexa546 + Cyt1Aa-Alexa647 (Fig 5C; *P* value < 0.001, Kruskal-Wallis test).

## Cry11Aa and Cyt1Aa proteins arranged into specific nanoscopic patterns at the apical region of the cell and in the cytosol

To address the nanoscopic location of Cry11Aa inside the cells and the observed colocalization of Cry11Aa and Cyt1Aa proteins during their synergistic interaction in the gut, we made use of super resolution radial fluctuation (SRRF) analysis, since this microscopy imaging technique overcomes the diffraction limits of confocal microscopy, providing nanoscopic information of fluorescent labeled structures [30]. First, we analyzed the images of larvae fed with Cry11Aa-Alexa546 inclusion. Highly brilliant cells after intoxication with Cry11Aa showing internalized Cry11Aa-Alexa546 protein were compared with the dark cells, which correspond to cells where the toxin was bound to the microvilli but was not internalized (Fig 4A and 4C). High-resolution analyses revealed that, besides being associated with the apical microvilli (Fig

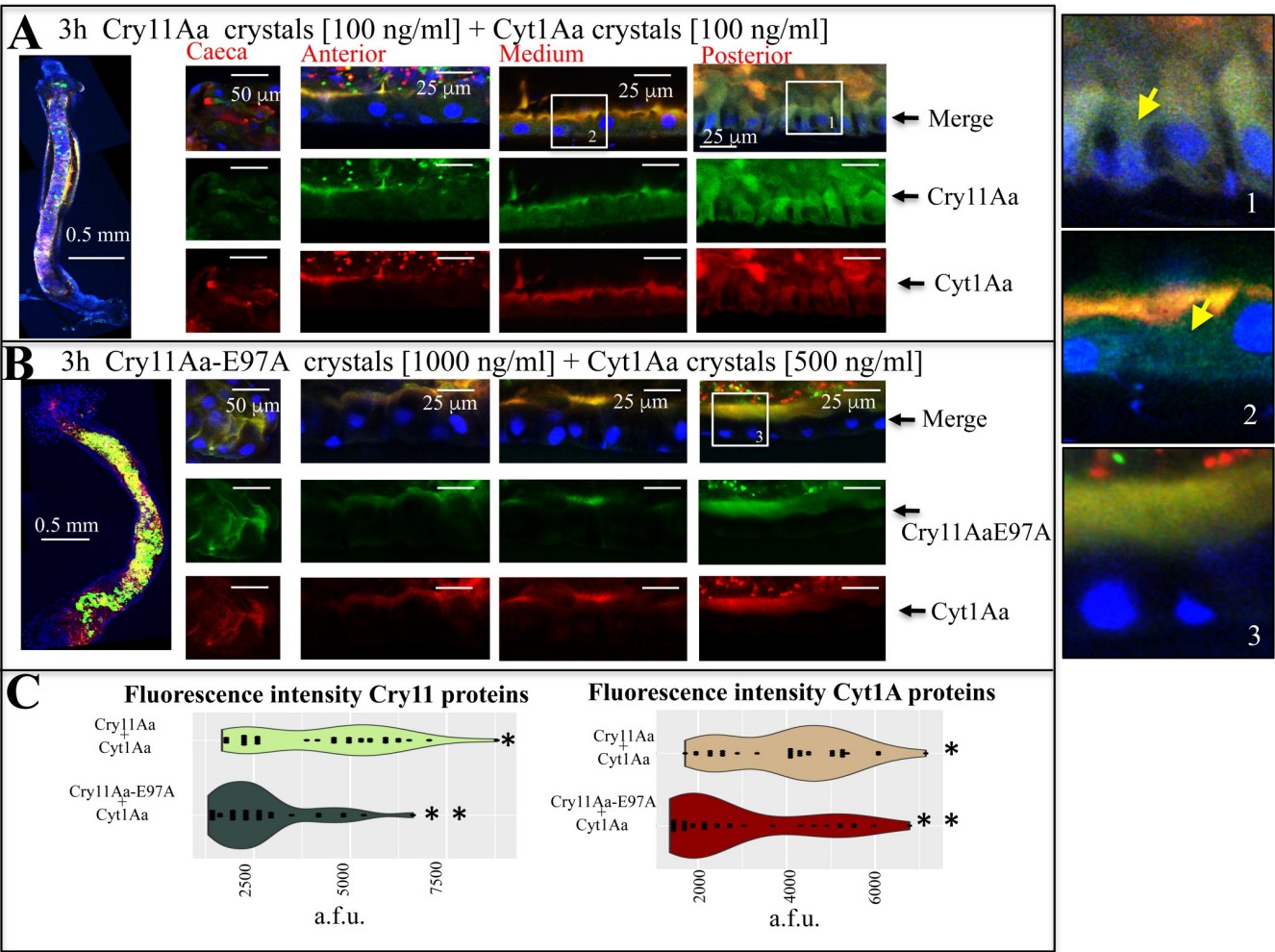

**Fig 5. *In vivo* co-localization of labeled Cry11Aa and Cyt1Aa proteins during their synergism in the midgut tissue of *Aedes aegypti* larvae fed with different mixtures of the labeled crystal inclusions.** A total of five larvae were dissected for each condition and each assay was performed at least three times. Representative images are shown. Panel A shows mixtures of 1:1 (100 ng/ml) crystal inclusions Cry11Aa-Alexa546 (green fluorescence) with Cyt1Aa-Alexa647 (red fluorescence). Panel B shows Cry11AaE97A-Alexa546 mutant (1000 ng/ml) with Cyt1Aa-Alexa647 (500 ng/ml). Larvae were fed with labeled crystal inclusions proteins for 3 h. Next, larvae were processed, nucleus was stained with DAPI (blue fluorescence) and labeled proteins were observed with confocal laser Olympus FV1000 scanning microscope. Insets show selected images to improve clarity of the images and yellow arrowheads point to highly brilliant midgut cells due to Cry11Aa internalization Panel C shows the intensity of fluorescence measurements of 20 x 20 μm squares selected from the cytoplasm of different cells. Data are shown as arbitrary fluorescent units (a.f.u.). Different asterisks indicate statistically significant different data analyzed by using Kruskal-Wallis test ($P$ value < 0.001).

6A and insets), the Cry11Aa-Alexa546 protein was internalized in structures displaying a "net like" organization within the whole cell cytoplasm (Fig 6A and insets). In contrast, the non-toxic Cry11Aa-E97A-Alexa546 mutant protein was only associated to the apical membrane (Fig 6B and insets).

To determine whether such "net-like" *xy* pattern organization appear to be clustered or randomly distributed, we scrutinized the cellular milieu of dozens of cells by means of computing the Ripley's K~ function [31]. The Ripley's K~ function computes the degree of organization of a distribution of fluorescent points by assessing the density of information of other fluorescent points at all radial distances [30]. Our data revealed a remarkable degree of organization of labeled Cry11Aa within the cytoplasm of Cry11Aa-brilliant cells. The K~ function computed over the Cry11Aa-Alexa546 distribution indicated that this protein accumulated inside

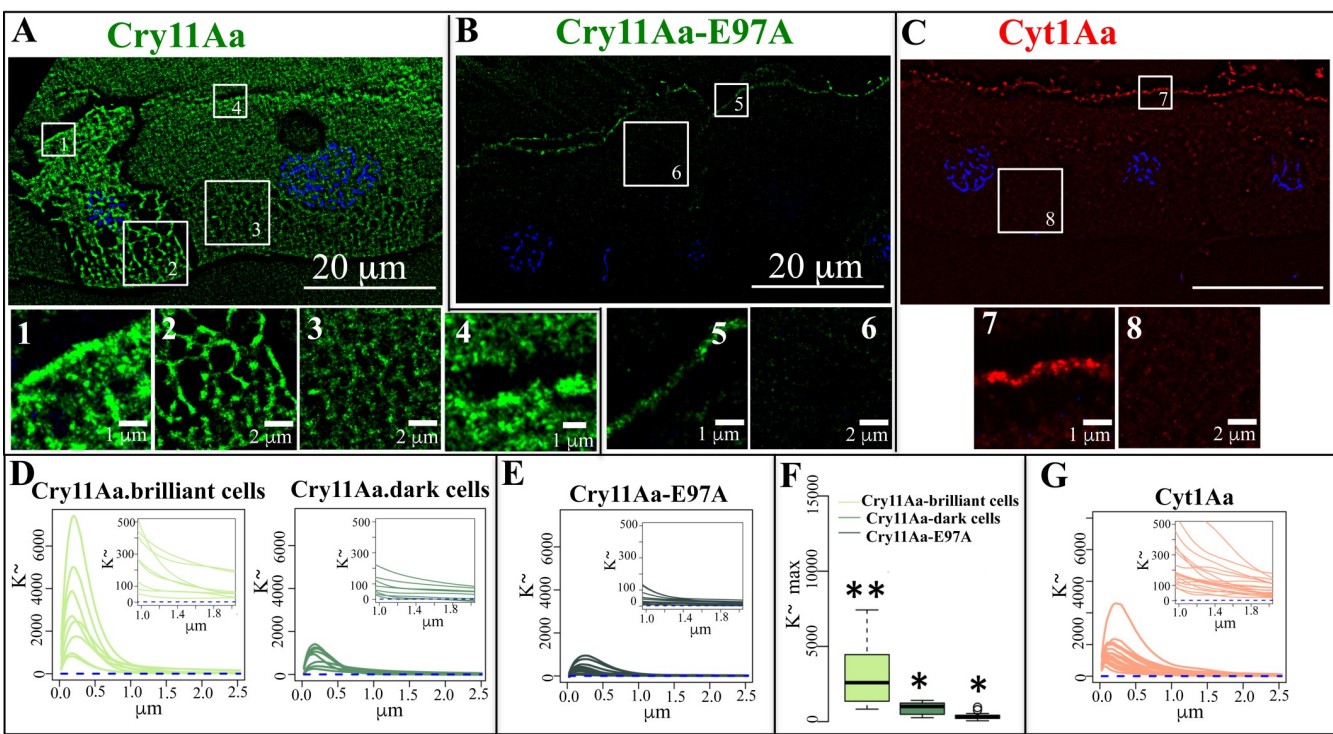

**Fig 6.** High resolution SRRF analysis showing the *in vivo* nanoscale localization of the labeled Cry11Aa-Alexa546 (Panel A), mutant Cry11AaE97A-Alexa546 (Panel B) and Cyt1Aa-Alexa647 (Panel C), in the midgut tissue of *Aedes aegypti* larvae. Cry11Aa proteins were labeled with Alexa546 showing green fluorescence. Individuals were fed with 1000 ng/ml of each labeled crystal inclusions proteins for 6 h. Next, larvae were processed, nucleus were stained with DAPI (blue fluorescence) and 100 frames of raw spinning disk data were recorded with the confocal Yokogawa spinning disk microscope alternating the laser lines illumination (405, 561 and 640 nm) per-frame basis, and Images were analyzed using NanoJ-core and NanoJ-SRRF plugins of Fiji-Image J software as described in Materials and methods. Panel A, shows one brilliant and one dark cell after Cry11Aa-Alexa546 ingestion. Insets show selected images to improve clarity of the images. Panel D, shows the Ripley's K-function statistical image analysis of the *xy* pattern organization observed in selected 10 x 10 μm squares selected from different cells of larvae intoxicated with Cry11Aa-Alexa546. Inset, shows an enlargement of the distal organization. The Cry11Aa-brilliant cells, and Cry11Aa-dark cells were analyzed. Similar analyses were performed for different cells of larvae intoxicated with Cry11AaE97A-Alexa546 (Panel E) or Cyt1Aa-Alexa647 (Panel G). These Ripley's K-function statistical image analysis compute the average number of particles located within a predefined radius of any typical event, normalized for the event intensity (density). Panel F, shows the Kruskal-Wallis test performed to the distributions with Cry11Aa-Alexa546 or Cry11AaE97A-Alexa546. Different asterisks indicate significant different data (*P* value = 0.05).

the cells in an ordered fashion (Fig 6A, 6D and inset -2). Such organization was over represented at the submicron scale (maximal values were found between 0.1 to 0.3 μm) (Fig 6D), but also scales down to the micron scale (1–2 μm) were observed (insets of Fig 6D). Such organization was much less prominent in the Cry11Aa-dark cells (Fig 6D), or when the larvae were fed with the mutant Cry11Aa-E97A-Alexa546 (Fig 6E). These observations are strengthened by comparing the distribution of K~ maxima values of Cry11Aa-brilliant cells with the corresponding computed values of either Cry11Aa-dark cells or Cry11Aa-E97A-Alexa546 protein distributions (*P*-value < 0.05, multiple comparisons after Kruskal-Wallis test) (Fig 6F).

The high-resolution images of cells intoxicated with Cyt1Aa-Alexa647 showed that this protein was associated with the apical membrane (Fig 6C and inset 7) and that its internalization was moderate (Fig 6C, 6G and inset 8) compared to the internalization of labeled Cry11Aa (Fig 6A and 6D and inset 2).

## Cry11Aa and Cyt1Aa distribution during synergism exhibited a dynamic nanoscopic organization layered at the apical membrane of the cell

It was reported that the synergism between Cry11Aa and Cyt1Aa toxins involves a high affinity interaction facilitating Cry11Aa oligomerization, increasing its pore formation activity [17,18].

Fig 5A shows that Cry11Aa and Cyt1Aa proteins colocalize at the microvilli membranes, hence supporting a putative synergistic interaction that might involve molecular contact. However, image resolution by confocal microscopy is limited by diffraction, to 0.2–0.3 μm. As result of that, the observed colocalization in the confocal microscopy, prevents us to derive conclusions at the nanoscale level.

To determine whether the observed colocalization (Fig 5A) when 4th instar larvae were fed 3 h with a 1:1 mixture of crystals from both proteins (100 ng/ml of each protein) is due to a precise arrange at the apical membrane of the cell, we analyzed their distribution by super resolution SRRF microscopy [30]. Fig 7A show that these proteins are layered into specific foci,

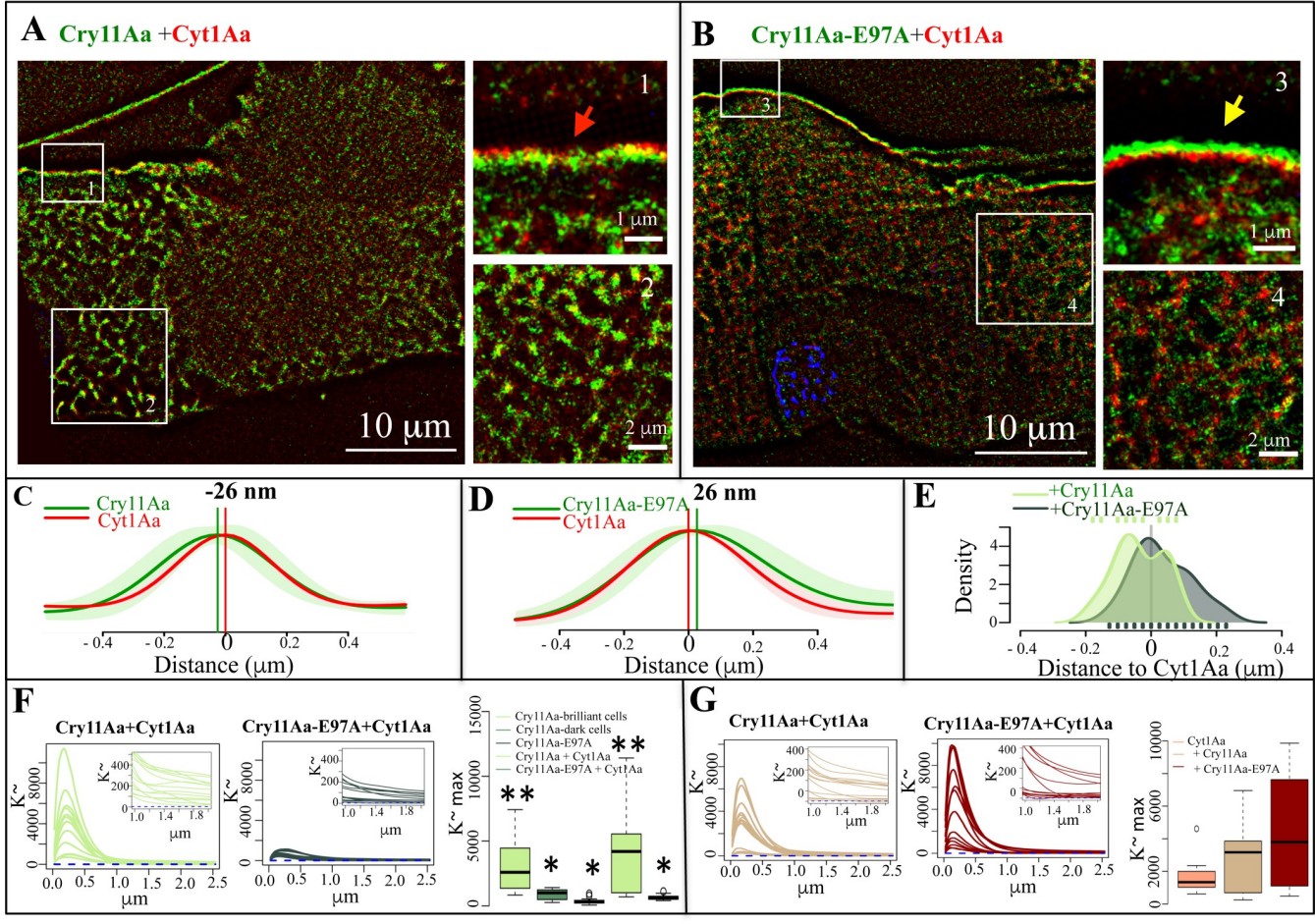

**Fig 7. Nanoscopic *in vivo* organization of Cry11Aa or Cyt1Aa proteins during their synergism in the midgut tissue of *Aedes aegypti* larvae.** Images were analyzed by high resolution SRRF microscopy. Larvae were fed 3 h with mixtures of crystal inclusions Cry11Aa-Alexa546 with Cyt1Aa-Alexa647 at 100 ng/mL of each protein (Panel A), and mutant Cry11AaE97A-Alexa546 with Cyt1Aa-Alexa647 at 1000 ng/mL + 500 ng/mL, respectively (Panel B). Larvae were processed as explained in Fig 6. Cry11Aa proteins were labeled with Alexa546 showing green fluorescence. Cyt1Aa proteins were labeled with Alexa647, showing red fluorescence. Panel A, shows one complete cell and one cell that is already broken down. Insets show selected images to improve clarity. Red arrowhead points to the specific Cyt1Aa-Alexa647 and Cry11Aa-Alexa546 proteins organization in apical membrane, where Cry11Aa-Alexa546 was located down facing the cell cytoplasm and Cyt1Aa-Alexa647 was found in the upper layer. Yellow arrowhead points to the inverted organization presented by Cry11AaE97A-Alexa546 mutant and Cyt1Aa-Alexa647. Panels C and D, show the quantitative line pattern analysis, where Cyt1Aa protein was arbitrary located at zero value, as described in Materials and methods, from more than 24 rectangles of 2.5 x 5 μm selected from Cry11Aa-Alexa546 with Cyt1Aa-Alexa647 mixture (Panel C), or from Cry11AaE97A-Alexa546 with Cyt1Aa-Alexa647 mixture (Panel D). Panel E, shows the distribution of the maximal distances of Cry11Aa and Cry11AaE97A to Cyt1Aa, which was arbitrarily located at zero value. Panels F and G, show the Ripley's K-function statistical image analysis of the *xy* pattern organization observed in 14 selected 10 x 10 μm squares selected from different cells of the larvae intoxicated with the different mixtures of proteins. Panel F, shows the analysis of the fluorescence of Cry11Aa-Alexa546 or mutant Cry11AaE97A-Alexa546 in these mixtures and Panel G, shows the analysis of the fluorescence of Cyt1Aa-Alexa647 in these mixtures. Panels F and G, also show the Kruskal-Wallis test analyses of these distributions, different asterisks indicate significant different data (*P* value = 0.05), and no asterisks indicate any statistical differences.

positioned on the external region of the apical membrane of the cell. Cyt1Aa-Alexa647 was found in the upper layer, while Cry11Aa-Alexa546 was located down facing the cell cytoplasm (Fig 7A, see red arrowhead in inset-1). Surprisingly, the analysis of the interaction of the mutant Cry11Aa-E97A-Alexa546 (1000 ng/ml) with Cyt1Aa-Alexa647, (500 ng/ml) after 3 h of intoxication, showed an inverted layered organization on the cell microvilli, where Cry11Aa-E97A-Alexa546 protein was found external to Cyt1Aa-Alexa647 at the apical membrane of the cell (Fig 7B, see yellow arrowhead in inset-3). Cry11Aa-E97A was previously shown to be unable to oligomerize and insert into liposomes which correlated with its severely impairment in insecticidal activity [26], indicating that the order array of Cry11Aa and Cyt1Aa in the membrane depends on the capacity of Cry11Aa to form oligomers and insert into the membrane.

To further quantitate the degree of organization at the apical membrane of cells from treated larvae, we scrutinized the relative distances of both protein-layered distributions. Our initial approximation was to compute line profiles over each protein distribution at the microvilli, and later to compare their proximity by looking at their local maxima. However, stretching and folding of both protein distributions was observed at the nanoscale regimes, hence preventing us to reach statistical significance. To overcome such limitation, we developed an algorithm that allowed us to quantitate the local distance between both protein layers, as described in Materials and methods. The algorithm stems on taking Cyt1Aa-Alexa647 distribution as a reference layer to assess its local degree of deformation computing its spatial paired correlation function, from dozens of regions located in the cell microvilli.

The data showed that both wild type toxins (Cry11Aa-Alexa546 and Cyt1Aa-Alexa647) were located at an average distance between each other of $\approx$ 26 nm (Fig 7C). The average distance between Cry11Aa-E97A-Alexa546 mutant with the Cyt1Aa-Alexa647 protein is also $\approx$ 26 nm, but in the opposite direction (Fig 7D). Fig 7E shows the distribution of all computed distances between Cry11Aa-Alexa546 or Cry11Aa-E97A-Alexa546 to Cyt1Aa-Alexa647 (where Cyt1Aa is positioned as the zero reference). Note that the distribution of distances between Cry11Aa-Alexa546 and Cyt1Aa-Alexa647 is shifted to the left side of Cyt1Aa (negative values), but the distribution of distances between Cry11Aa-E97A-Alexa546 and Cyt1Aa-Alexa647 is shifted to the right side of Cyt1Aa (positive values) (Fig 7E). Kruskal-Wallis test performed over the distributions of Fig 7E indicate statistical difference ($P$ value < 0.0001). Noteworthy, the distribution of all computed distances between Cry11Aa-Alexa546 or Cry11Aa-E97A-Alexa546 to Cyt1Aa-Alexa647 are rather broad, both covering positive and negative values (Fig 7E). Hence, indicating that the interaction between these proteins is rather dynamic, where Cry11Aa and Cyt1Aa toxins may have different distances between them. The location of the wild type Cry11Aa protein showed two peaks (Fig 7E), suggesting that this protein may be found in two different conformations, one colocalizing over the Cyt1Aa, and other under the Cyt1Aa facing the cytoplasm, while the mutant Cry11Aa-E97A that is unable to oligomerize could be found colocalizing with Cyt1Aa in only one conformation on the top of the Cyt1Aa protein.

Our super-resolution studies also showed that the wild type Cry11Aa-Alexa546 protein was found inside the cells (Fig 7A) during synergism with Cyt1Aa, similar to the organization pattern observed when larvae were fed only with Cry11Aa-Alexa546 crystals (Fig 6A). Statistical image analysis of the organization of both toxins showed that wild type Cry11Aa-Alexa546 had much higher K~ values than the mutant Cry11Aa-E97A-Alexa546 (Fig 7F), indicating internalization of Cry11Aa-Alexa546 in an ordered close and distal conformations. The Kruskal-Wallis test over the distributions of K~ maxima computed values from the wild type Cry11Aa-Alexa546 protein against Cry11Aa-E97A-Alexa546 K~ maxima values, revealed statistically significant differences ($P$ value < 0.05) (Fig 7F). These data confirmed our

observation that the mutant is not capable of internalization into the cell cytoplasm, even though the concentration of Cry11Aa-E97A-Alexa546 was 10 times fold higher than Cry11Aa-Alexa546 in these comparisons.

From the confocal images shown in Fig 5B in relation to the interaction of Cyt1Aa-Alexa647 with Cry11Aa-E97A-Alexa546, no apparent internalization of Cyt1Aa-Alexa647 inside cells was observed (Fig 5B). However, the super resolution images revealed some Cyt1Aa-Alexa647 inside the cell (Fig 7). The analysis of the distribution of Cyt1Aa-Alexa647 inside the cells when it was fed to the larvae in a mixture with Cry11Aa or in a mixture with Cry11Aa-E97A protein showed similar patterns (Fig 7G), and Kruskal-Wallis test confirmed that these patterns showed no significant differences.

## Discussion

It has been documented that insects are able to regulate their feeding behavior when exposed to toxins [32]. In the case of different lepidopteran insect species, it has been shown that larvae selectively choose diet that does not contain Bt Cry toxins [33–35]. In case of mosquito larvae, it was reported that filtration rates of three different mosquito species decreases significant after ingestion of Bti crystals [36]. Specifically, it was reported that *Ae. aegypti* stop feeding one h after ingestion of lethal dose of Bti crystals [36]. Here, we show that when *Ae. aegypti* larvae were fed toxic soluble pore-forming toxins of Cry11Aa or Cyt1Aa the larvae survived, and these labeled proteins were barely detected inside the gut. In contrast, when larvae were fed with labeled soluble non-toxic Cry11Aa-E97A and Cyt1Aa-V122E mutants, these two labeled proteins were observed to accumulate in their gut lumen and bound to microvilli membranes. Ingestion rates in the presence of soluble active toxin or mutant toxins remains to be analyzed in the future. As mentioned in introduction, the particle size that was estimated to be retained in the mosquito midgut was from 0.7 to 26 μm [5]. Thus not surprisingly, when larvae were fed with crystal inclusions of Cry11Aa, Cyt1Aa or combination of crystals from both proteins, these proteins accumulated inside the gut despite their high toxicity. A possible explanation could be that, since crystals inclusions require to be solubilized under the alkaline pH of mid-gut lumen and activated by the proteases present in the gut [1], crystal inclusions may mask the toxicity of the insecticidal proteins for a short period allowing ingestion of high quantities of Bt proteins by the larvae. It is possible that slow solubilization of these proteins *in vivo* allowed the ingestion of substantial amount of toxic crystals before the larvae could sense toxicity and stop feeding. It was proposed that the transit time of Cry and Cyt crystals along the mosquito larvae gut is from 30–60 min, depending on species [37]. We also observed that the Cry and Cyt inclusions were always inside the gut lumen suggesting that they remain in the mosquito gut lumen as inclusion bodies for an extended periods of time.

Both Cry11Aa and Cyt1Aa are recognized as pore forming toxins and in the case of Cyt1Aa it was also proposed that it may induce a detergent like mechanism. The observations made here shows that both toxins exert toxicity by different mechanisms. It was reported that Cry11Aa toxin associates preferentially to microvilli from *caeca* and posterior midgut, which correlates with the sites where Cry11Aa receptors (ALP, APN and CAD) were shown to be preferentially expressed [22–25]. However, our data show that Cry11Aa or the non-toxic Cry11AaE97A mutant also binds to the microvilli of the anterior and medium midgut regions. Interestingly, it was recently shown that CRISPR-Cas9 edited *Ae. aegypti* larvae expressed the CAD, fused to the green fluorescent protein (cadEGFP) principally in the *caeca* and posterior midgut regions, but also low expression of this protein was observed in anterior-medium midgut region [25]. Thus, CAD could participate in Cry11Aa binding to anterior and medium

midgut. In the future, it will be important to determine if additional Cry11Aa-receptor molecules are involved in the binding of Cry11Aa toxin to anterior and medium gut regions.

The non-toxic Cry11Aa-E97A mutant used in this work is in helix α-3 of domain I. We previously showed by circular dichroism analysis, that this mutant retains the structure of Cry11Aa toxin. However, this mutation affects oligomerization in synthetic membranes, suggesting there is no pore formation activity. But its binding characteristics to BBM isolated from mosquito larvae were not described previously [26]. A similar corresponding mutation in the lepidopteran-specific Cry1Ab protein, Cry1Ab-R99E mutant affected oligomerization, pore formation and toxicity against *Manduca sexta* larvae. But the mutant was still able to bind to its receptors, with similar affinity as Cry1Ab [38]. Here we show that Cry11Aa-E97A mutant, although is not toxic, it is still also able to bind BBM of mosquito larvae.

We show that when larvae were fed Cry11Aa-Alexa546 crystal inclusions at low doses (100 ng of crystals/ml) the toxin was bound to the microvilli of *caeca* and posterior region. Some microvilli shedding was observed at 24 h, as previously described [25]. It was proposed that a rapid loss of the cell microvilli containing the toxic pores of Cry11Aa could be a defense mechanism of the larvae to prevent further damage [25]. However, here we also observed internalization of Cry11Aa in certain cells of the midgut. At higher dose (1000 ng of crystals/ml) the number of cells that showed internalization of Cry11Aa increased, and at longer incubation times these cells were completely degraded. In contrast, Cyt1Aa showed a strong association with the microvilli from all gut regions, which is compatible with the detergent like mechanism proposed previously for Cyt1Aa [8,13]. Our data show that this protein was not accumulated to high levels inside cells and did not induce a clear destruction of the cells during intoxication. These data are interesting since we have previously reported that in the mosquito *Culex quinquefasciatus* the Cyt1Aa toxin also binds to microvilli from all midgut regions, but it also can be observed inside cells, internalized on small vesicles in the cells of *caeca* and anterior midgut regions, although this distribution pattern requires further analysis [39].

An additional striking difference between both Cry11Aa and Cyt1Aa was the massive degradation of the *caeca* gut structure, induced by Cry11Aa but not by Cyt1Aa, even though both toxins were shown to bind to *caeca* microvilli. The complete destruction of the *caeca* structure by Cry11Aa action was not previously reported, and it was remarkable that when larvae were fed with both proteins, Cry11Aa and Cyt1Aa, this effect was enhanced. It is possible that internalization of Cry11Aa into cells is responsible for the degradation of the gut structures. Cry11Aa cell internalization in the *caeca* was more difficult to observe, mainly due to the complex structure of *caeca*, that does not allow clear transversal sections. Interestingly, we show that Cry11Aa cell internalization was triggered by its pore formation activity, since Cry11Aa-E97A mutant, was not internalized into the cells.

Different Cry toxins have been shown to bind actin, and silencing actin expression in *Ae. aegypti* larvae caused increased toxicity of Cry11Aa, supporting a possible defense mechanism [40–43]. Interestingly, Cry5Ba that is toxic against nematodes, is internalized inside gut cells in small vesicles by RAB-5 endocytosis as a defense mechanism, involving microvilli shedding and destruction of Cry5Ba pores after vesicles fusion with the lysosomes [44]. In our opinion this internalization of Cry5Ba in small vesicles is more similar to the internalization that we observed with the soluble Cry11Aa protoxin at 9 and 24 h of intoxication, probably representing also a defense strategy against protoxin action, since all larvae survived after treatment with soluble protoxin. It has been shown that other pore-forming toxins cause release of cell blebs containing cytoplasm material, which may result from the engagement of cell death pathways triggered by pore formation activity [45,46]. It remains to be determined the reason for the high Cry11A cell internalization, as well as to identify the Cry11Aa internalization pathway.

There are multiple studies showing the histopathological effects of Cry toxins in midgut cells observed by light or transmission electron microscopy in different insect orders including lepidopteran [47–52], coleopteran [53] or dipteran larvae [54–57]. Overall, these reports show that Cry toxins induced severe vacuolization of the cytoplasm of the midgut cells, as well as loss of microvilli structure, cell lysis and columnar cells fragmentation. It was even proposed that vacuolization resulted from increased Golgi vesicles, up to 4 μm in size [49]. Some authors have immunolocalized the toxin in infected larvae. However, none of them observed Cry toxin internalization into cells, only that the toxin binds to the microvilli membrane [47,52,55]. One possible explanation for these different results, is that in the analysis of fixed tissues, antigen unmasking is needed, to allow the interaction of the antibodies with their epitopes, and trypsin treatment of the tissue sections during unmasking likely damages the tissue [47,52,55]. Another explanation could be that only part of the toxin is internalized into the cells and antibodies used to immunolocalize the protein did not recognize the cleaved fragment of the protein. Here, we used proteins labeled with fluorescent dyes that bind to Lys residues, which are found all over the Cry11Aa and Cyt1Aa proteins.

Finally, our high-resolution microscopy analyses revealed that Cyt1Aa and Cry11Aa have a dynamic ordered layered pattern of organization in the membrane microvilli, which is compatible with the mechanism where Cyt1Aa functions as a receptor of Cry11Aa facilitating its oligomerization, and inducing its membrane insertion. This distribution of Cyt1Aa and Cry11Aa layers on the microvilli could only be observed by using high resolution microscopy such as the SRRF technique. Interestingly, high-resolution images revealed that Cry11Aa was finally localized facing cell cytoplasm, below the Cyt1Aa in the microvilli. In contrast, the interaction of Cry11Aa-E97A with Cyt1Aa revealed an inverted organization pattern in the microvilli membrane. It has been shown that after membrane insertion, the Cry1Ab oligomers are mobilized into lipid membrane rafts where the toxin inserts and forms pores [58]. Thus, the different organization in the microvilli membrane of the mutant Cry11Aa-E97A is likely due to its defects in oligomerization and membrane insertion.

Overall, our results allowed us to break down some paradigms that were proposed before to explain the mode of action of these toxins. The mode of action of Cry11Aa and Cyt1Aa greatly differs and we show that the mechanism of Cry11Aa is more complex than previously proposed, involving other effects besides pore-formation. It would be fundamental to determine the intracellular events that are triggered after toxin cell internalization. Also, there is a need to identify the Cry11Aa receptors in anterior and medium regions of the larvae. Decoding the complete mechanism of Cry and Cyt toxins is likely to provide future strategies for their efficient use in pest control, and for the control of insects that develop resistance to these toxins.

## Materials and methods

### Toxins preparation

Cyt1Aa, Cyt1Aa-V122E, Cry11Aa and Cry11Aa-E97A proteins were produced in the *B. thuringiensis* 407 acrystalliferous strain transformed with the corresponding expression plasmids, codifying separately for each toxin protein such as pWF45-Cyt1Aa or pWF45-Cyt1Aa-V122E [27,59] and pCG6-Cry11Aa or pCG6-Cry11Aa-E97A [26,60]. The construction and characterization of the non-toxic mutants (Cyt1Aa-V122E and Cry11Aa-E97A), affected in oligomerization and insertion into membrane, was previously described [26,27]. Bacteria cultures were grown in solid nutrient broth sporulation medium [61] supplemented with erythromycin (10 μg/mL) for 72 h until sporulation (≥80%), detected in an optical microscope. Spores/crystals were recovered and washed once with acidic distilled water adjusted to pH 4 using 5 M HCl, and then three times with 0.3 M NaCl, 0.01 M EDTA, pH 8.0 and three times with 1 mM

phenylmethylsulfonyl fluoride (PMSF) and stored at 4˚C. Cry11Aa and Cry11Aa-E97A crystal inclusions were purified by sucrose gradient centrifugation [62], while Cyt1Aa and Cyt1Aa-V122E inclusions were purified by the aqueous two-phase system composed of phosphate buffer 40% and polyethylene glycol (PEG) 40% as previously described [12,63]. Cry11Aa and Cry11Aa-E97A protoxins were solubilized in 50 mM $Na_2CO_3$, 0.2% β-mercaptoethanol, pH 10.5 and centrifuged for 10 min at $12,850 \times g$, 4˚C. Cyt1Aa and Cyt1Aa-V122E were solubilized in 50 mM NaOH, 1 mM dithiothreitol (DTT), 1 mM PMSF and centrifuged for 10 min at $12,850 \times g$, 4˚C. The soluble protoxins were recovered in the supernatant. For protoxin activation we first adjusted the pH to ~8.5 by adding ¼ volume of 1 M Tris HCl pH 8.0. Protoxins were incubated with trypsin from bovine pancreas (Sigma-Aldrich, St Louis, MO, USA) in a mass ratio of 1:20 w/w (1 h, at 37˚C). A final concentration 1 mM of PMSF was added to stop proteolysis and proteins were dialyzed against 0.02 M $Na_2HPO_4$ buffer, pH 8, at 4˚C, for 16 h. Bradford assay were used to determine protein concentration (Biorad, Hercules, CA, USA), and protein integrity was analyzed through SDS-PAGE. Molecular weight markers were Precision Plus Protein Standards All Blue (Bio-Rad) and masses are indicated in kDa (S2 Fig).

## Bioassays

*Ae. aegypti* were reared for more than 15 years at Instituto de Biotecnología (UNAM) facilities, at 28˚C, 75% humidity, with a 12 h: 12 h, light: dark, photoperiod. Larvae were fed commercial cat food and adults artificially fed with cow blood treated with 0.5 mg/ml heparin (Sigma) set in a Petri dish covered with a Parafilm™ membrane (Sigma-Aldrich, St Louis MO). Groups of five 3[rd] instar larvae in quintuplicate were placed in 2 ml of dechlorinated water using 24-well tissue culture plates (1.76 cm diameter/well) and food was not administrated during the bioassay. Larvae were treated with different toxin protein concentrations (100 to 10,000 ng/ml, determined in Bradford assays) of spore/crystal suspensions, soluble protoxins, or soluble activated toxins. A mixture of Cry11Aa and Cyt1Aa proteins (1:1 ratio) was also analyzed at the same concentrations using crystal inclusions, protoxins or activated toxins as indicated in the text. Negative control (dechlorinated water) was included in the bioassay. Mortality was recorded after 24 h. Each bioassay was repeated three times and the medium concentration to kill 50% of the exposed larvae ($LC_{50}$) was estimated by Probit analysis using Polo Plus Probit and Logit Analysis version 1.0 LeOra software. The synergism factor was determined according to Tabashnik's equation that assumes a simple additive effect [28] by dividing the theoretical toxicity by the observed toxicity of the bioassays performed with 1:1 mixtures of Cry11Aa and Cyt1Aa proteins.

## Confocal microscopy

The soluble Cry11Aa protoxins and activated toxins, or purified Cry11Aa crystal inclusions were labeled with the fluorescent dye Alexa546 succinimidyl ester (NHS) that label Lys residues (S2 Fig) according to the manufacturer's instructions (Thermo Fisher Scientific, Waltham, USA). The Cyt1Aa proteins (protoxins or crystal inclusions) were labeled with the fluorescent dye Alexa647-NHS. The efficiency of labeling was measured using the molar extinction coefficient of each probe and the following equation:

$$\frac{A_x}{\varepsilon} \times \frac{\text{MW of protein}}{\text{mg protein/ml}} = \frac{\text{moles of dye}}{\text{moles of protein}} \tag{Eq 1}$$

where: $A_x$, corresponds to the maximum wavelength of dye absorbance, 554 nm for Alexa Fluor-546 and 650 nm for Alexa Fluor-647; and ε, corresponds to the molar extinction coefficient of Alexa Fluor-546 that is 203,000 $M^{-1}$ $cm^{-1}$ at 554 nm and for Alexa Fluor-647 is 270,000

$M^{-1}$ $cm^{-1}$ at 650 nm. Purity and integrity of these proteins was analyzed on SDS-PAGE with 15% acrylamide. The labeling of the proteins was visualized by excitation of the SDS-PAGE gel with Epi-RGB light transilluminator and analyzed in an Amersham Imager 600 (GE, Irving, TX).

Second instar *Ae. aegypti* larvae were fed several days with cooked oats (25 mg/L boiled 5 min in water), to avoid auto fluorescence of the standard rearing diet (cat food). When larvae reached early 4th instar, they were placed into 24-well tissue culture plates at room temperature containing 100 ng/ml or 1000 ng/ml of labeled proteins Cry11Aa or Cyt1Aa (soluble proteins or crystal inclusions) for different exposure times (3, 6, 9 and 24 h) in the dark. In the 1:1 mixture of Cyt1Aa with Cry11Aa crystal inclusions, 100 ng/ml of each labeled inclusions were used, while mixture of inclusions of Cry11Aa-E97A with Cyt1Aa was used at 1000 ng/ml and 500 ng/ml, respectively. The midgut tissue was dissected from the larvae at the indicated times, the food bolus retained in the peritrophic membrane was not removed and these tissues were fixed overnight in 4% paraformaldehyde supplemented with 5% sucrose in PBS pH 7.4. These midgut tissues were carefully washed three times with PBS and labeled with 4',6-diamidino-2-fenilindol (DAPI) (blue fluorescence) (Sigma-Aldrich, Saint Louis, MO) at 5 mg/ml, final concentration in PBS, for 30 min. After DAPI labeling, the tissues were washed again with PBS, placed on microscopic slides and covered with Prolong Glass antifade mountant (Invitrogen, Thermo Fisher Scientific, Waltham, USA). We used two strips of masking tape in each side of the slide glass to protect the tissue from being pressed by the cover glass. Proteins labeled with Alexa546 were excited at 543 nm and fluorescence emission was analyzed at 568 nm (green fluorescence). Alexa647-conjugated proteins were excited at 633 nm and fluorescence emission was recorded at 665 nm (red fluorescence). Images were obtained using confocal laser scanning microscope (Olympus FV1000) at the Laboratorio Nacional de Microscopía Avanzada located in the Instituto de Biotecnología-UNAM facilities. All figures showing the whole gut are composite images constructed with Fiji-Image J software (using BigStitcher plugin) [64]. Individual images used for composite images are shown in S8–S11 Figs. A total of five larvae were dissected for each condition and each assay was performed at least three times. The images in the figures are representative images of the obtained results.

## Analysis of the fluorescence intensity

The intensity of fluorescence of labeled proteins at the cytoplasm region of infected gut cells was quantified from confocal images as a proxy of protein abundance. Squares of 20 x 20 μm covering a region of interest (ROI) within the cytoplasmic region were selected from different cells (14–30 cells per experimental treatment). Reported intensity fluorescence values were computed with image J software [64]. Statistical analyses were performed with multiple comparison test after Kruskal-Wallis. Differences with *P* value < 0.001 were considered as statistically significant.

## Super-resolution radial fluctuation (SRRF) microscopy

Confocal images were collected with a spinning disk confocal unit (Yokogawa CSU-W1) mounted on an epifluorescence microscope (Zeiss Observer Z.1). Detection of the signal was via a 60X S objective, numerical aperture: 1.3 (oil immersion) and an electron multiplying-CCD camera (Andor Ixon 3 EMCCD, model DU-897E-CS0-BV, Andor Technology) configured on a full chip mode. Images were collected at 2.4 sec intervals using 3i proprietary TTL synchronization electronics and SlideBook software. Fluorescence excitation was performed by alternating the laser lines illumination (561, 405 and 640 nm) on a per-frame basis with 0.2,

0.4 and 0.5 sec exposure times, respectively, and camera gain parameters (x300). Sub-diffraction images were derived from the SRRF approach [30,65].

SRRF is a super resolution microscopy technique that overcomes the diffraction limit of light by analyzing the statistical properties of a sequence of images collected from the same imaging plane. In this study we collected 100 confocal images at the same imaging plane. Each super-resolved image is derived from analyses of blinking and bleaching statistics of the staining fluorophores. In the present study samples where fixed with 4% paraformaldehyde supplemented with 5% sucrose in PBS pH 7.4, hence this temporal image stack of 100 confocal images harbors, at single pixel level, dynamics of the photophysical transitions experienced by fluorophores due to the fluorescence excitation regime. The temporal analysis of such transitions (explained below) is required to be analyzed in order to overcome the diffraction limit of light, hence, to increase the resolution of the final reconstructed image.

The serial stacks of the 100 images were analyzed using the NanoJ-core and NanoJ-SRRF plugins of Image J [64]. The SRRF-approach encompasses two steps of image processing. First, each diffraction-limited image (i.e. a confocal image) is subjected to a spatial analysis, where information about the of local changes on the fluorescence field is used to shape an image with increased resolution. These images are called radiality maps (RM) because each contains information that points to centers of convergence of the fluorescent field (the fluorophores). Starting with a temporal stack of 100 confocal images collected over the same scene will give 100 RMs with increased resolution.

In the second step, the stacks of RMs are subject to a temporal analysis that further increases resolutions and damper imaging artifacts. The RMs were analyzed using the SOFI approach [66], which increases resolution through analyzing the temporal fluorescence fluctuations of the fluorophores. Finally, the stack of 100 frames of RMs were integrated on a super-resolution image.

Settings of NanoJ-SRRF plugin [64] were as follows, ring radius 0.5, radiality magnification 10, and axes in ring 8 parameters. For the temporal analyses, the second order cumulant of the temporal auto-correlation function (TRAC2) option was activated on the advanced settings of NanoJ-SRRF plugin.

## Quantification of nanoscopic protein distributions at apical membranes

To study the relative distances of the protein distributions, ROIs of 5 μm x 10 μm (34 rectangles from each condition) were selected from their super-resolution images of apical membranes. Each ROI was manually rotated at ImageJ, using bicubic interpolation, to finally set the signal of both fluorescence channels at the horizontal, i.e. the fluorescence of Cry11Aa and Cyt1Aa proteins distributions at the apical membrane orientated along to the horizontal axis. These images were analyzed as intensity carpets, using the R statistical software, with columns representing nanoscopic replicates ($\partial r$ = 25 nm width) of line profiles of the fluorescence intensity distribution of each protein layer.

Stretching and folding of both protein distributions was observed at the nanoscale regimes. Hence, intensity line profiles carpets were aligned, by computing their paired correlation function:

$$pCF(y, \partial r) = \frac{< F(y,0)F(y + \partial y, \partial r) >}{< F(y,0) >< F(y, \partial r) >} - 1 \tag{Eq 2}$$

With $F(y, \partial r)$ representing a line intensity profile of the fluorescence distribution of Cyt1Aa-Alexa647 computed at the $y$ axis, which was defined to be perpendicular to the apical membrane (at the $y$ axis). $F(y, 0)$ corresponds to the reference intensity profile of the same

carpet. Notation $< \cdots >$ indicate the expected value, i.e. $<F>$ represents the mean value of the fluorescence intensity. The argument of the maxima of the pCF, arg max($pCF$), was then used to identify the shifts between local maxima of the line profiles of a given carpet, each separated at a $\partial r$ distance from the reference line profile. Those shifts were used to align both Cyt1Aa-Alexa647 ($F_1$), and Cry11Aa-Alexa546 ($F_2$) florescence carpets or Cyt1Aa-Alexa647 ($F_1$), and Cry11Aa-E97E-Alexa546 ($F_2$) florescence carpets. An average representation of hundreds of line profiles aligned accordingly, including up to 10 cellular replicates of each experimental condition, is shown in Fig 7C and 7D. Confidence intervals represent standard error of the mean estimated at the level of cellular replicates.

The local distance between both Cyt1Aa and Cry11Aa protein layers was assessed from their corresponding paired cross correlation function:

$$pCCF(y, \partial r) = \frac{< F_1(y, 0)F_2(y + \partial y, \partial r) >}{< F_1(y, 0) >< F_2(y, \partial r) >} - 1 \qquad \text{(Eq 3)}$$

The argument of the maxima of the pCCF, arg max($pCCF$), was then used to identify distances between local maxima.

## Spatial statistics

Each data associated with a protein in a super-resolution microscopy image has a coordinate $(x, y)$ attached to it, to make a pattern of points, providing valuable information about the variables and processes collected in these points. Thus the characterization of a dot pattern inside the cells is therefore of interest for the description of some intra-cellular processes and could be carried out through spatial analysis statistics of the super-resolution microscopy images.

The spatial analysis implemented in this work seek discrimination between the three types of point structures: i) concentrated, with a high density of points in specific areas of the image; ii) scattered, the points tend to occupy most of the study area maximizing the distance between points, and (iii) established randomly, there is no pattern at the points in the image.

In this work, we analyzed the organization of Bt toxins within the larvae gut by means of using a variant of the Ripley's K-function, which provides a mathematical framework to quantitatively assess the degree of organization of the fluorescence field of a super-resolved image.

If a group of points is distributed randomly, by example using a Poisson process with density $\lambda$, the expected number of points in a circle radius $d$ is $\lambda \pi d^2$. Deviations from randomness can be quantified by the Ripley's K-function, which reflect the type, intensity and range of the spatial pattern by analyzing the existing distances between all points. In general terms, the Ripley's K-function can be computed as the average number of particles located within a predefined radius, which is normalized for the intensity (density) over the the entire image, including all potential particle locations.

We made use of a modified Ripley K function proposed by Amgad *et al.* [31], which allow the identification of overlapping events at extremely high event densities, an approach suitable to quantitate protein aggregation in fluorescent microscopic images [31]. The $K$ metrics proposed by Amgad *et al.* [31] encompass the Besag´s correction [67], one of the best methods to reduce bias of the K function near the edges of the image. This correction is needed since it is not possible to assess the organization of the data near the boundaries of the image due to the lack of information outside of the image. Besag's correction implements a theory to compensate such bias by proposing a normalization equation that considers proximity to the edges of the image [67].

For testing complete randomness we used an implementation of Ripley's K-function that normalizes for the event density and have unit variance [68]. Once the $\tilde{K}$ function is computed, it is compared to its upper and lower boundary limits, (typically, the 1st and 99th

quantiles), as proposed by Lagache et al. [64] to assess if a value would be in a complete spatial randomness, or not.

Interpretations about the meaning of the $\tilde{K}$ metrics were as follows:

*i*, When the observed $\tilde{K}$ value is greater than the confidence interval (compare curves with dashed lines in Figs 6D, 6E, 6G, 7F and 7G), the spatial clustering at that distance is statistically significant.

*ii*. When the observed $\tilde{K}$ value is lower than the confidence interval, the spatial dispersion at that distance is statistically significant.

Calculations of $\tilde{K}$ were made over the super-resolution images using a MATLAB script provided by Amgad *et al.* [31] (MATLAB R2020b). At least 14 squares of 10 x 10 μm from the cytoplasm of the super-resolution images of different cells of each condition were analyzed. ROIs super-resolution images were separated in single channels, exported in TIF format (8-bits, gray scale images) and used as input element for the MATLAB script. The $\tilde{K}$ metric was computed for each fluorescence channel separately at different radii *r* at steps of $\Delta r = 25$ nm, with their respective critical quantiles (Figs 6D, 6E, 6G, 7F and 7G). Statistical analyses were performed over the arg $\max(\tilde{K})$ considering multiple comparison test after Kruskal-Wallis (Figs 6F, 7F and 7G). A *P* value < 0.05 was considered as statistically significant.

## Supporting information

**S1 Fig. Description of the mosquito midgut tissue.** Panel A, Schematic drawing of the mosquito larva food channel. Panel B, Image of the midgut tissue dissected from *Aedes aegypti* larva, highlighting the main regions of the midgut. Abbreviations: Ph, pharynx; CG, gastric *caeca*; pm, peritrophic membrane; lc, central midgut lumen; AMG, anterior midgut; CMG, medium central midgut; PMG, posterior midgut; MT, Malpighian tubules: Rc, rectum; Ac, anal canal; FG, Forward midgut; MG, Midgut; and HG, hindgut.
(TIF)

**S2 Fig. SDS-PAGE protein profile of labeled Cry11Aa and Cyt1Aa proteins.** Panel A, Labeled Cry11Aa proteins with fluorescent dyes Alexa546 (green color) and Cyt1Aa with Alexa647 (red color) were separated in SDS-PAGE 10% and the gel was stained with Coomassie Blue or visualized under Epi-RGB light. Molecular markers (kDa) are on the left. Panel B, Pure labeled Cry11Aa-Alexa546 and Cyt1Aa-Alexa647 crystals visualized in the confocal laser-scanning microscope excited at 543 nm and analyzed at 568 nm fluorescence emission (Alexa546) or excited at 633 nm and recorded at 665 nm (Alexa647). Clear field microscopy images of the crystals are also shown in this figure.
(TIF)

**S3 Fig. *In vivo* localization of soluble activated toxins from Cry11Aa-Alexa546 or Cry11AaE97A-Alexa546 mutant proteins, in the midgut tissue of *Aedes aegypti* larvae.** Individuals were fed with 1000 ng/ml of each labeled soluble activated toxin for 3, 6, 9 or 24 h. Midgut tissue was dissected and processed, as indicated in Materials and methods. Nucleus was stained with DAPI (blue color) and labeled proteins (green color) were observed with confocal laser Olympus FV1000 scanning microscope. Clear field microscopy images of *caeca* region in each condition are also shown in this figure. Red arrow head points to the binding of soluble Cry11Aa toxin to the microvilli membrane of the posterior midgut region and white arrowheads point to the binding of soluble Cry11Aa-E97A toxin to the microvilli membrane of the anterior and medium regions.
(TIF)

**S4 Fig.** *In vivo* **localization of soluble protoxins from Cry11Aa-Alexa546 or Cry11AaE97A-Alexa546 mutant proteins, in the midgut tissue of** *Aedes aegypti* **larvae.** Individuals were fed with 1000 ng/ml of each labeled soluble protoxin, for 3, 6, 9 or 24 h. Midgut tissue was dissected and processed, as indicated in in Materials and methods. Nucleus was stained with DAPI (blue color) and labeled proteins (green color) were observed with confocal laser Olympus FV1000 scanning microscope. Clear field microscopy images of *caeca* region in each condition are also shown in this figure. Inserts show selected images to improve clarity of the images, and orange arrowheads points to Cry11Aa protoxin that was found internalized in small vesicles into the cells of medium midgut region after 9 and 24 h.
(TIF)

**S5 Fig.** *In vivo* **localization of soluble protoxins of Cyt1Aa-Alexa647 or mutant Cyt1Aa-V122A-Alexa647, in the midgut tissue of** *Aedes aegypti* **larvae.** Individuals were fed with each labeled soluble protoxin (1000 ng/ml) for 3, 6, 9 or 24 h as indicated in the figure. Midgut tissue was dissected and processed, as indicated in Materials and methods. Nucleus was stained with DAPI (blue color) and labeled proteins (red color) were observed with confocal laser Olympus FV1000 scanning microscope. Clear field microscopy images of *caeca* region in each condition are also shown in this figure.
(TIF)

**S6 Fig.** *In vivo* **localization of crystals from Cyt1Aa-Alexa647 or Cyt1AaV122A-Alexa647 mutant proteins, in the midgut tissue of** *Aedes aegypti* **larvae.** Individuals were fed with labeled crystals (1000 ng/ml) for 3, 6, 9 or 24 h. Midgut tissue was dissected and processed as indicated, in Materials and methods. Nucleus was stained with DAPI (blue color) and labeled proteins (red color) were observed with confocal laser Olympus FV1000 scanning microscope. Clear field microscopy images of *caeca* region in each condition are also shown in this figure.
(TIF)

**S7 Fig.** *In vivo* **co-localization of Cry11AaE97A or Cyt1Aa proteins in the midgut tissue of** *Aedes aegypti* **larvae fed with crystals of the different proteins.** Larvae were fed with 1000 ng/ml of Cry11AaE97A-Alexa546 and 500 ng/ml of Cyt1Aa-Alexa647 labeled crystals for 9 h. Midgut tissue was dissected and processed, as indicated in Materials and methods. Nucleus was stained with DAPI (blue color), Alexa546 labeled proteins (green color) and Alexa647 labeled proteins (red color) were observed with confocal laser Olympus FV1000 scanning microscope.
(TIF)

**S8 Fig. Images used to construct composite images of** Fig 2**.**
(TIF)

**S9 Fig. Images used to construct composite images of** Fig 3**.**
(TIF)

**S10 Fig. Images used to construct composite images of** Fig 4**.**
(TIF)

**S11 Fig. Images used to construct composite images of** Fig 5**.**
(TIF)

## Acknowledgments

The authors thank the teams from insectarium of IBT-UNAM (Lizbeth Cabrera) and LNMA-UNAM personal (Andres Saralegui, Xochitl Alvarado and Arturo Pimentel) for their technical support.

## Author Contributions

**Conceptualization:** Maria Helena Neves Lobo Silva-Filha, Sarjeet S. Gill, Mario Soberón, Alejandra Bravo.

**Formal analysis:** Adán Guerrero.

**Funding acquisition:** Sabino Pacheco, Sarjeet S. Gill, Mario Soberón, Alejandra Bravo.

**Investigation:** Samira López-Molina, Nathaly Alexandre do Nascimento, Sabino Pacheco.

**Methodology:** Jorge Sánchez.

**Resources:** Sarjeet S. Gill, Mario Soberón.

**Software:** Adán Guerrero.

**Supervision:** Mario Soberón, Alejandra Bravo.

**Validation:** Adán Guerrero, Alejandra Bravo.

**Visualization:** Alejandra Bravo.

**Writing – original draft:** Alejandra Bravo.

**Writing – review & editing:** Maria Helena Neves Lobo Silva-Filha, Adán Guerrero, Sarjeet S. Gill, Mario Soberón, Alejandra Bravo.

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
