## [Decision Letter · Decision Letter 0]

14 Oct 2020

Dear Dr. Bravo,

Thank you very much for submitting your manuscript "In vivo nanoscale analysis of the dynamic synergistic interaction of Bacillus thuringiensisCry11Aa and Cyt1Aa toxins in Aedes aegypti." for consideration at PLOS Pathogens. As with all papers reviewed by the journal, your manuscript was reviewed by members of the editorial board and by two independent reviewers. In light of the reviews (below this email), we would like to invite the resubmission of a significantly-revised version that takes into account the reviewers' comments.

Both reviewers felt the core observations derived from the high resolution microscopy studies are of potential value to the field.  However, as discussed in detail by Reviewer 2, much of the text describing the results is confusing, illogical, and/or suffers from substandard language usage.  For further consideration by the journal, a complete rewrite that addresses these shortcomings is required.

Please see also the attached annotated copy of the manuscript on which Reviewer 1 has added a number of comments that require your attention.

We cannot make any decision about publication until we have seen the revised manuscript and your response to the reviewers' comments. Your revised manuscript is also likely to be sent to reviewers for further evaluation.

Sincerely,

Michael R. Wessels

Section Editor

PLOS Pathogens

Kasturi Haldar

Editor-in-Chief

PLOS Pathogens

orcid.org/0000-0001-5065-158X

Michael Malim

Editor-in-Chief

PLOS Pathogens

orcid.org/0000-0002-7699-2064

Reviewer's Responses to Questions

**Part I - Summary**

Reviewer #1: This comprehensive manuscript (highly significant) refines a conventional paradigm explaining the mode of action of mosquitocidal proteins Cry11Aa and Cyt1Aa. Several proposed remarks that would improve the manuscript are included as sticky notes throughout the text in the attached MS.

Reviewer #2: This manuscript uses very high-resolution light microscopy with labelled proteins to investigate the spatial orientation of Bt proteins in the mosquito larva in vivo. This is a new application of this powerful technique in the field of Bacillus thuringiensis. However, presentation of results before the microscopy is confusing and contradictory, unnecessarily so. Misleading statements are made that have nothing to do with the insights from microscopy. If these are cleaned up, this would be a valuable contribution to the field.

**Part II – Major Issues: Key Experiments Required for Acceptance**

Reviewer #1: None required.

Reviewer #2: Abstract, lines 30-33 are misleading. 1) soluble proteins are in fact toxic. 2) no evidence that larvae stop feeding for any reason is given. 3) whether larvae ingest nontoxic proteins has no relation to the first two points. See discussion of 1) below.

"Only crystal inclusions were toxic, which remain insoluble in the gut-lumen for a prolonged time." The authors are implying that solubility is detoxifying the crystal, which is incorrect.

Information on the size of the crystals is necessary. What size of particle is trapped by the mosquito larva's filter-feeding apparatus, and what size of particle is not?

line 131. The authors should add the information from Ref 21 that soluble proteins attached to latex beads were trapped by the mosquito larvae and killed them. Soluble proteins are toxic.

The observation is not "reinforced" by the fact that mosquito larvae are filter feeders. It is explained by that fact.

The authors state that soluble material is excluded from the gut. This is incorrect--it is not excluded from the gut. The concentration of pre-solubilized proteins in the gut can be no greater than its concentration in the medium. Thus the proteins are toxic but not at that concentration. The authors would need to measure the concentration of protein recently dissolved from trapped crystals in the midgut to know what the toxicity of soluble proteins is.

The statements that crystals are toxic and soluble proteins are not toxic are not correct. A crystal that does not form soluble pore-forming proteins cannot kill the insect. Crystals per se are not toxic. Soluble proteins form pores and kill the insect. Soluble proteins are toxic. The authors are skipping over the filter-feeding aspect of mosquito larva that causes particle-size-dependent concentration and this has led them to make statements that are descriptions of the phenomena of what happens when crystals or previously solubilized proteins are put in the water. However these are not accurate statements about toxicitity. The data in the manuscript are valid, but the interpretation is unnecessarily confusing.

More explanation of Figure 2 is required. Exactly what region corresponds to the caeca? The anterior midgut? the posterior midgut? are there other regions not named? Is there a food bolus or was it removed? Is there a peritrophic matrix or was it removed? There needs to be a diagram showing the regions mentioned in the text. Since some images were constructed from composites, the originals must be presented in the Supplementary Material. What is the dark region in the middle? Is it a food bolus surrounded by a peritrophic matrix? Is Figure 2B just showing movement of the food bolus? In Figure 2D, comment on the appearance of the midgut after 9 hours as well as the caeca. Does degradation of the caeca cause a connection between the gut lumen and the body cavity, or does the caeca just shrink in size?

lines 183-184. Labelling proteins does not cause a mutation. Do the authors want to compare mutant proteins with wild-type proteins, or labelled proteins with unlabelled proteins?

lines 189-195. The data do not support the authors' assertion that larvae did not ingest soluble toxin. The data at 3 hours show that they did ingest it, but that they later got rid of it or somehow changed the membrane to make it less adhesive to the toxin.

The authors need to set their results in the context of quantitative measurements on how much the filter-feeding apparatus increases the concentration of particles actually ingested, relative to the concentration of particles in the water. There must be data available on that somewhere, for some mosquito species. This should be a function of particle size, or at least minimum particle size, and time (because more particles will accumulate in the apparatus for longer times).

lines 217-221. This paragraph does not make sense. The authors claim that wild-type Cry11Aa which could be toxic, is not toxic because it is not ingested, and the mutant Cry11Aa-E97 is ingested because it could not be toxic. This does not support the statement "the lack of toxicity of Cry11Aa soluble proteins correlated with a strong reduction of their ingestion." This explanation does not make sense. Is the mosquito doing something to get rid of the wild-type protein that it is not doing to the mutant protein? And how repeatable is the observation in Figure 3A?

what is the principle of the labelling with the Alexa dyes? "According to manufacturer's instructions" is not specific enough. Many different functional groups could be targetted. When a crystal is labelled, how much of the protein is labelled? Is only the protein on the outside of the crystal labelled?

line 374, precludes us to derive conclusions > prevents us from making conclusions. The words "precludes", "precluding" are used incorrectly throughout the manuscript.

lines 246-247 the arbitrary designation P1 and P2 for cells without or with label internalization is useless. Don't add to the abbreviations the reader must keep in mind to understand the paper.

lines 267-274. I do not understand the point of this paragraph and Figure S4. Why is it surprising that labelled crystals can be seen in the lumen? Why is it interesting that they get washed out or dissolved over time? The authors state "the crystal inclusions may spend a long period of time inside the larval gut", but most of them do not.

line 278, 289-290 independent evidence, such as behavioral observations, is required to conclude that the mosquito larvae stopped feeding. It cannot be concluded with the data presented here.

**Part III – Minor Issues: Editorial and Data Presentation Modifications**

Reviewer #1: See the sticky notes in the attached MS.

Reviewer #2: line 70. "particularity" is not a word in English. In general, there are several errors in English which I would have expected to be corrected by the coauthor with the greatest familiarity with the language.

For each figure showing something labelled with a color, the description of all of the colors and what they label must be in the caption for that figure. Red, green, and blue.

The SRRF Microscopy is not a standard technique, and the authors have tried to explain the methods. But the point of using this technique is lost in the details. The authors should explain the method in terms understandable to a scientist not familiar with the technique, in such a way as to explain why it is being used.

line 687 why temporal analysis? What is changing over time?

line 754 state in words what the K-function is supposed to measure, and what it tells us about the distribution of particles. The purpose of the technique is getting lost in the description of the method.

line 755-757 a diagram would be helpful here.

line 768 please give an intuitive explanation of why edge effects disrupt the K-function calculation, and how Besag's correction solves this problem.

lines 783-789 this discription is helpful. But what does it mean for the image analysis? Are we only interested in clustered distributions? If so, why?

line 314, the word "awry" is never used as an adjective in English.

The abbreviation IF is defined once. This is a useless abbreviation. Use the words "fluorescence intensity" the first time, and the word "fluorescence" subsequently.

Line 346. What are "tents of cells"?

line 403. is 26 nm the distance between the two proteins, or the distance between the two labels on the two proteins? Does this tell us anything about the orientation of the proteins?

line 441 it is not clear what comparisons are described here. what pattern is not significantly different from what other pattern?

line 451-452. The statement of reduced ingestion can only be supported by behavioral observations, not the evidence given in the manuscript.

line 466 If the authors are tempted to propose such an unreasonable hypothesis, they must state how it could be tested. Or, more likely, that it cannot be tested, given that many other Bt toxins are found in crystals of protoxins, which are consumed by lepidoptera or coleoptera.

line 525. The authors cite Ref. 39 which shows that RAB-5 mediated endocytosis defends nematodes against Cry5Ba damage. Could the RAB-11-dependent shedding of microvilli shown in the same paper be responsible for the destruction of the caecum?

line 539 the explanation of the "net" structure within the cytoplasm is not convincing.

The orientation of Cyt1Aa and Cry11Aa layers on the microvilli is very interesting. The authors should explicitly state that it could only be observed using the SRRF technique.

PLOS authors have the option to publish the peer review history of their article (what does this mean?). If published, this will include your full peer review and any attached files.

Reviewer #1: No

Reviewer #2: No
---

## [Decision Letter · Decision Letter 1]

30 Nov 2020

Dear Dr. Bravo,

We are pleased to inform you that your manuscript 'In vivo nanoscale analysis of the dynamic synergistic interaction of Bacillus thuringiensisCry11Aa and Cyt1Aa toxins in Aedes aegypti.' has been provisionally accepted for publication in PLOS Pathogens.

Best regards,

Michael Wessels

Section Editor

PLOS Pathogens

Kasturi Haldar

Editor-in-Chief

PLOS Pathogens

orcid.org/0000-0001-5065-158X

Michael Malim

Editor-in-Chief

PLOS Pathogens

orcid.org/0000-0002-7699-2064

Reviewer Comments (if any, and for reference):

Reviewer's Responses to Questions

**Part I - Summary**

Reviewer #1: This comprehensive manuscript (highly significant) refines a conventional paradigm explaining the mode of action of mosquitocidal proteins Cry11Aa and Cyt1Aa.

Reviewer #2: The authors have carefully addressed most of my criticisms, and I agree completely with their responses and alterations to the manuscript.

**Part II – Major Issues: Key Experiments Required for Acceptance**

Reviewer #1: (No Response)

Reviewer #2: (No Response)

**Part III – Minor Issues: Editorial and Data Presentation Modifications**

Reviewer #1: (No Response)

Reviewer #2: (No Response)

PLOS authors have the option to publish the peer review history of their article (what does this mean?). If published, this will include your full peer review and any attached files.

Reviewer #1: No

Reviewer #2: No

---

## [Editor Report · Acceptance letter]

12 Jan 2021

Dear Dr. Bravo,

We are delighted to inform you that your manuscript, "In vivo nanoscale analysis of the dynamic synergistic interaction of Bacillus thuringiensisCry11Aa and Cyt1Aa toxins in Aedes aegypti.," has been formally accepted for publication in PLOS Pathogens.

Best regards,

Kasturi Haldar

Editor-in-Chief

PLOS Pathogens

orcid.org/0000-0001-5065-158X

Michael Malim

Editor-in-Chief

PLOS Pathogens

orcid.org/0000-0002-7699-2064